# Combined Effect of Rarefaction and Effective Viscosity on Micro-Elasto-Aerodynamic Lubrication Performance of Gas Microbearings

**DOI:** 10.3390/mi10100657

**Published:** 2019-09-29

**Authors:** Yao Wu, Lihua Yang, Tengfei Xu, Haoliang Xu

**Affiliations:** 1State Key Laboratory for Strength and Vibration of Mechanical Structures, School of Aerospace Engineering, Xi’an Jiaotong University, Xi’an 710049, China; nealjackman@stu.xjtu.edu.cn (Y.W.); xtf1992@stu.xjtu.edu.cn (T.X.); xuhaoliang@stu.xjtu.edu.cn (H.X.); 2Shaanxi Key Laboratory of Environment and Control for Flight Vehicle, School of Aerospace, Xi’an Jiaotong University, Xi’an 710049, China; 3School of Aerospace Engineering, Xi’an Jiaotong University, Xi’an 710049, China

**Keywords:** gas rarefaction, micro-elasto-aerodynamic lubrication, compliance matrix, effective viscosity, bearing characteristics

## Abstract

Elastic deformation and gaseous rarefaction effects are of great importance to the static and dynamic characteristics of gas microbearings. Based on the effective viscosity model of Veijola, the governing equations can be solved by the partial derivative method, finite element procedure, and relaxed iterative algorithm. The numerical results showed that the maximum gas pressure is relatively lower compared to a microbearing with a rigid liner at a local pressure peak region, owing to the film thickness of two converging-diverging profiles and the existence of bimodal pressure inside the elastic microbearing liner. However, the effect of bearing flexibility provides a marginal increase in the load capacity on account of the integral area of pressure distribution is larger than the rigid bearing liner. The friction coefficient and direct stiffness coefficients increase as the elastic modulus decreases while the direct damping coefficients become smaller at high eccentricity ratios and bearing numbers. Since the Poiseuille flow rate increases in connection with an increasing Knudsen number, the effective viscosity of the lubricant leads to a decreased load carrying capacity, friction coefficient, and direct stiffness coefficient, which produces an increase in the direct damping coefficients.

## 1. Introduction

For applications involving micro-electromechanical systems (MEMS) microfluidic devices with high power efficiency and density, such as micro gas turbines, microgenerators, micromotors, medical devices, and hard disk drives (HDDs), there has been great interest in micro gas journal bearings. Previous investigations show that self-acting gas bearings offer certain advantages over the rolling element bearings, magnetic bearings, and oil/water-lubricated bearings including simpler structure design, higher precision, lower power loss, lower frictional characteristics, no contamination due to leakage of lubricants and lesser restrictions on speed and temperature [1,2,3,4,5]. In the conventional bearing design of hydrodynamic lubrication, it has been assumed that the bearing shell is rigid and the lubricant viscosity is constant. When the geometric parameters of the gas slider bearing are given, the bearing number (Sommerfeld number) is only related to the journal speed. These assumptions are unrealistic especially for the microbearings operating with ultra-thin lubrication films. As the film thickness becomes thinner, it is possible that the elastic deformation of bearing bounding surface alter the clearance space geometry of the bearing. Based on the Veijola’s effective viscosity formula [6,7], the compressible lubricant viscosity is influenced by the Knudsen number *K_n_*, which characterizes the degree of gas rarefaction. The Knudsen number is defined as the ratio of mean free path of gas molecules *λ*_0_ to the characteristic length scale of the gas film [8,9]. Under such conditions, in order to guarantee the safety of micro rotating machines, the coupling effect between the effective gas viscosity, gaseous rarefaction, and compressibility, as well as the flexibility of bearing shell should be investigated for better understanding the micro flow mechanisms in ultra-thin film elastoaerodynamic lubrication.

Generally, the gas flow characteristics in microdevices differ from those in the macrodevice and the flow behaviors deviate dramatically from continuum flows. The ultra-thin gas film lubrication between the magnetic head and disk assembly has been studied both experimentally and analytically. Renowned authors of the literature include Burgdorfer [10], Hsia [11], Mitsuya [12], Fukui, and Kaneko [13,14]. They derived the modified Reynolds equations by incorporating their different Poiseuille flow factors for arbitrary Knudsen numbers. Since Dowson [15] originally developed the solution of hydrodynamic lubrication for highly loaded elastic cylinders in the 1950s, a substantial amount of tribological research has been devoted to the elastohydrodynamic lubrication (EHL). Some studies have examined the effect of bearing liner deformation on the slider bearing characteristics (e.g., the pressure distribution, load carrying capacity, and frictional force) [16,17,18,19,20,21,22] and they indicated that the elastic parameter could strongly affect the friction coefficient and the flow rate. Others focused on the impact of the journal eccentricity perturbation or the journal misalignment and the rheological effects of couple stress fluids on stiffness and dynamic coefficients of finite journal bearing with the elastic bearing shell, such as gas foil bearing, offset-halves pressure dam bearing, coupled journal-thrust bearing [23,24,25,26,27,28,29]. Another aspect of the research of elastic bearing properties is that of viscoelasticity. Zhang et al. [30,31] utilized the finite difference method for determining the static characteristics of gas journal microbearings including the effect of rarefaction and the effective viscosity of the lubricant. The load-carrying capacity was found to decrease considerably with an increase in the Knudsen number. Karadere and Gultekin [32] adopted the constitutive equations of the linear elastic materials to model the pad and runner deformations under stable lubricant viscosity and isothermal conditions and compared the solutions of steel runner–steel pad and the steel runner–bronze pad material pairs. The elastohydrodynamic numerical simulation for conformal contacts of fixed slider plane bearings was made by Yagi and Sugimura [33]. The pressure produced in the film and the minimum friction coefficient was appreciably affected by the piezoviscous effect and elastic deformation. Su et al. [34] applied Boussinesq’s semi-infinite body theory to estimate the elastic deformation of surface textured sliding bearings with relatively soft material, they concluded that small elastic deformation, texture area density, and viscosity-pressure effect of the lubricant decrease the bearing load capacity at the thin-film hydrodynamic lubrication. Chetti and Boualem [35] introduced Constantinescu’s turbulent theory for the elastohydrodynamic lubrication of journal bearings lubricated with couple stress fluids. The combined stiffness and damping models of oil films established by Zhou et al. [36] for the line contact EHL of the gear drive. The small normal and tangential stiffness related to effective viscosity, entrainment velocity, and shear rate are obtained along the line of action. Schapery [37,38] published the findings of the developmental work for bearing sizing computer programs based on the linear elasticity equations and the assumption of rigid shims. A parameterized calculation model of hybrid journal bearings with compliant polymer liner was developed by Linjamaa et al. [39], which emphasizes the importance of elastic and thermal deformation in the operational performance. However, as the fluid film thickness decreases, there are very few numerical results, which give the rarefaction effects within ultra-thin gaseous films on elastoaerodynamic problem. Besides the effects of gas rarefaction, most literature on EHL analysis have generally been carried out for the steady characteristics of journal bearing and thin film lubrication data for dynamic coefficients of micro gas bearings are scarcely investigated. Its micro lubricating mechanism is still limited because of the complicated aeroelastic behavior. In addition, the viscosity of rarefied gas flow needs to be considered along with the change in the Knudsen number, high computational cost, and poor convergence would be expected in the solution of micro-elasto-aerodynamic lubrication. Therefore, it is essential to elucidate the interaction effects of rarefaction, effective viscosity, and elastic deformation of the bearing surface on the operation of gas microbearings in engineering practices.

The combined effect of rarefaction, bearing shell flexibility and effective viscosity are taken into account for the flow characteristics of the gas-lubricated journal microbearings in micro-fluidic systems. The modified Reynolds equation incorporating the Poiseuille flow rate for rarefied gas are solved simultaneously with three-dimensional elasticity equations to predict the film pressure distribution and the elastic deformation of the bearing liner; the results are compared with those obtained from the rigid microbearing, which ignores the effective viscosity in ultra-thin gas film lubrication. Important information, such as the dynamic characteristics in microbearing with an elastic liner and the effective viscosity of the gas lubricant is of practical importance in the bearing design to improve the microbearing-rotor system stability.

## 2. Governing Equations

Schematic diagrams of a gas journal microbearing are shown in Figure 1. In current analysis, inertia effects and thermal effects in the lubricating film are neglected and the rarefied gas flow is laminar. Owing to the elastic modulus of journal material is much larger than that of the bearing bush material, the rotor shaft is assumed to be rigid. The modified Reynolds equation including rarefaction and bearing liner elastic deformation for determining pressure distribution within the entire domain can be written in a non-dimensional form as:(1)$$\frac{\partial }{\partial \varphi }\left(QPH^{3}\frac{\partial P}{\partial \varphi }\right)+\frac{\partial }{\partial \lambda }\left(QPH^{3}\frac{\partial P}{\partial \lambda }\right)=\Lambda_{eff}\frac{\partial (PH)}{\partial \varphi }+2\Lambda_{eff}\frac{\partial (PH)}{\partial T}$$
where *P* = *p*/*p_a_*, *H* = *h*/*c*, *φ* = *x*/*R*, and *λ* = *z*/*R* are the dimensionless gas pressure, dimensionless local film thickness, coordinate in the slider length direction, and coordinate in the slider width direction. *p_a_* is the ambient pressure, *c* is the radius clearance, *R* is the journal radius, *p* is the local gas pressure, *h* is the local film thickness taking into account the elastic deformation of the bearing liner, and *ε* is the eccentricity ratio. Λ*_eff_* = 6*μ_eff_ωR*^2^/(*p_a_c*^2^) is the effective bearing number, *μ_eff_* is the effective viscosity of the lubricant. *ω* is the rotating angular velocity of the journal and *T* is the dimensionless time.

For more accurate prediction of microbearing characteristics, the effective gas viscosity *μ_eff_* of gas lubricant should be used to amend the static gas viscosity *μ*. Cercignani [40] and Saraf [41] developed the solutions for Kramers’ problem, plane Couette flow, and plane Poiseuille flow using the variational method. These studies indicate that the stress in the rarefied gas flow regions appears as:(2)$$\tau_{xz}=\left(\frac{1}{2}+T_{1}(D)+\frac{\sqrt{\pi }}{2}J_{\min}\right)\tau_{xz,fm}$$
where *D* is the inverse Knudsen number, $D=\sqrt{\pi }/(2K_{n})$. *T_n_*(*x*) is an Abramowitz function defined by [42]:(3)$$T_{n}(x)=\int_{0}^{\infty }t^{n}\cdot e^{-t^{2}-\frac{x}{t}}dt$$
(4)$$2T_{n}(x)=(n-1)T_{n-2}(x)+xT_{n-3}(x),\;n\geq 3$$
where *J*_min_ is the minimum value of the function *J(u)*.
(5)$$J(u)=\int_{-\frac{D}{2}}^{\frac{D}{2}}{\left[u(x)\right]}^{2}dx-\sqrt{\pi }\int_{-\frac{D}{2}}^{\frac{D}{2}}\int_{-\frac{D}{2}}^{\frac{D}{2}}T_{-1}(\left|x-y\right|)u(x)u(y)dxdy-2\int_{-\frac{D}{2}}^{\frac{D}{2}}u(x)dx$$
where *u*(*x*) is the mass velocity.

The expression for *J*_min_ in plane Poiseuille flow is:(6)$$J_{\min}=-\frac{c_{22}{c_{1}}^{2}-2c_{12}c_{1}c_{2}+c_{11}{c_{2}}^{2}}{c_{11}c_{22}-{c_{12}}^{2}}$$
where the coefficients *c*_1_, *c*_2_, *c*_11_, *c*_12_, and *c*_22_ is given by:(7)$$\left\{ \begin{array}{l} c_{1}=\frac{D^{3}}{12},c_{2}=D,\\ c_{11}=\sqrt{\pi }\left[\begin{array}{l} 8-\frac{\sqrt{\pi }}{12}D^{3}+\frac{D^{4}}{16}-2D\left(4+D^{2}\right)T_{0}(D)-\left(16+8D^{2}+\frac{D^{4}}{8}\right)T_{1}(D)-\\ D\left(16+D^{2}\right)T_{2}(D) \end{array}\right]\\ c_{12}=\sqrt{\pi }\left[2-\frac{\sqrt{\pi }}{2}D+\frac{D^{2}}{4}-2DT_{0}(D)+\left(4+\frac{D^{2}}{2}\right)T_{1}(D)-2DT_{2}(D)\right]\\ c_{22}=\sqrt{\pi }\left[1-2T_{1}(D)\right] \end{array}\right.$$

Following the above equations, Veijola [7] presented the effective viscosity model through curve fitting, which is a function of the Knudsen number:(8)$$\mu_{eff}=\frac{K_{n}}{1+2K_{n}+0.2K_{n}^{0.788}e^{-\frac{K_{n}}{10}}}\cdot \mu$$

*Q* is the relative Poiseuille flow rate coefficient and is defined as the ratio of Poiseuille flow rate *Q_P_* for rarefied gas flow to that for continuum flow *Q*_con_. The Poiseuille flow rate coefficient *Q_P_* with three adjustable coefficients and the flow rate coefficient of a continuum flow are expressed in terms of the inverse Knudsen number *D* as follows [43]:(9)$$Q_{P}=\frac{\frac{1}{2\mu }\frac{\partial p}{\partial x}h^{3}\left(b\cdot D^{c}+\frac{a\sqrt{\pi }}{2D}+\frac{1}{6}\right)}{\frac{-h^{3}}{2\mu D}\cdot \frac{\partial p}{\partial x}}=b\cdot D^{c+1}+\frac{a\sqrt{\pi }}{2}+\frac{D}{6},Q_{con}=\frac{D}{6}$$

The dimensionless Poiseuille flow rate ratio *Q* between journal and bearing bush surfaces for a wide range of Knudsen number can be written as:(10)$$Q=\frac{Q_{P}}{Q_{con}}=\frac{b\cdot D^{c+1}+\frac{a\sqrt{\pi }}{2}+\frac{D}{6}}{\frac{D}{6}}=1+\frac{3a\sqrt{\pi }}{D}+6b\cdot D^{c}$$

For three adjustable coefficients *a* = 0.01807, *b* = 1.35355 and *c* = −1.17468, the Poiseuille flow rate ratio [44] based on high-order boundary condition is simplified to:(11)$$Q=1+0.10842K_{n}+9.3593/{K_{n}}^{-1.17468}$$

Under steady-state conditions, the transient term ∂(*PH*)/∂*T* of Equation (1) can be ignored and the following form for modified Reynolds equation is expressed as:(12)$$\frac{\partial }{\partial \varphi }\left(QPH^{3}\frac{\partial P}{\partial \varphi }\right)+\frac{\partial }{\partial \lambda }\left(QPH^{3}\frac{\partial P}{\partial \lambda }\right)=\Lambda_{eff}\frac{\partial (PH)}{\partial \varphi }$$

To obtain the micro elastic aerodynamic lubrication performance of gas microbearing with compliant liner, the modified Reynolds equation including the effective viscosity and elastic deformation of the bearing liner should be solved numerically. It is difficult to solve the Equation (1) due to its first-order derivative terms in time, hence the partial derivative method [45,46] is adopted. After the mathematical transformation *PH* = *S*, (*PH*)^2^ = *S*^2^ = *Π*, equation (1) becomes an ellipse-type partial differential equation −∇⋅(c∇u)+au=f in the following form:(13)$$\begin{array}{l} -(\frac{\partial^{2}\Pi }{\partial \varphi^{2}}+\frac{\partial^{2}\Pi }{\partial \lambda^{2}})+\frac{2\Pi }{H}(\frac{\partial^{2}H}{\partial \varphi^{2}}+\frac{\partial^{2}H}{\partial \lambda^{2}})+\frac{2\Pi }{QH}(\frac{\partial Q}{\partial \varphi }\frac{\partial H}{\partial \varphi }+\frac{\partial Q}{\partial \lambda }\frac{\partial H}{\partial \lambda })=\\ -\frac{1}{H}(\frac{\partial H}{\partial \varphi }\frac{\partial \Pi }{\partial \varphi }+\frac{\partial H}{\partial \lambda }\frac{\partial \Pi }{\partial \lambda })+\frac{1}{Q}(\frac{\partial Q}{\partial \varphi }\frac{\partial \Pi }{\partial \varphi }+\frac{\partial Q}{\partial \lambda }\frac{\partial \Pi }{\partial \lambda })-\frac{2\Lambda_{eff}}{QH}\frac{\partial S}{\partial \varphi }-\frac{4\Lambda_{eff}}{QH}\frac{\partial S}{\partial T} \end{array}$$

The Equation (12) is solved for nondimensional pressure *P* subject to the boundary conditions:(14)$$\left\{ \begin{array}{l} P{|}_{\varphi ,\lambda =\pm \frac{B}{2R}}=1,\\ P{|}_{\varphi =0,\lambda }=P{|}_{\varphi =2\pi ,\lambda },\\ \frac{\partial P}{\partial \varphi }{|}_{\varphi =0,\lambda }=\frac{\partial P}{\partial \varphi }{|}_{\varphi =2\pi ,\lambda } \end{array}\right.$$
where *B* is the width of the bearing.

The components of load support at *x* and *y* coordinates are calculated by integrating the gas film pressure acting on the journal surface:(15)$$\left\{ \begin{array}{l} {\bar{F}}_{x}=p_{a}R^{2}\int_{-\frac{B}{2R}}^{\frac{B}{2R}}\int_{0}^{2\pi }(P-1)\sin\varphi d\varphi d\lambda \\ {\bar{F}}_{y}=p_{a}R^{2}\int_{-\frac{B}{2R}}^{\frac{B}{2R}}\int_{0}^{2\pi }(P-1)\cos\varphi d\varphi d\lambda \end{array}\right.$$

As a result, the nondimensional load-carrying capacity *C_L_* and the attitude angle *θ* are evaluated by:(16)$$C_{L}=\frac{W}{p_{a}RB}=\frac{R}{B}\int_{-\frac{B}{2R}}^{\frac{B}{2R}}\int_{0}^{2\pi }(P-1)\cos\varphi d\varphi d\lambda$$
(17)$$\theta =\arctan\left(\frac{{\bar{F}}_{x}}{{\bar{F}}_{y}}\right)$$

The frictional force in the dimensionless form *F_b_* on the journal surface can be computed by integrating the shear stress:(18)$$F_{b}=-\int_{-\frac{B}{2R}}^{\frac{B}{2R}}\int_{0}^{2\pi }\left(\frac{\Lambda_{eff}}{6}\frac{1}{H}+\frac{H}{2}\frac{\partial P}{\partial \varphi }\right)d\varphi d\lambda$$

The gas microbearing behaves like a spring and a damper in the bearing-rotor system. The linear small perturbation method is employed here to derive the dynamic lubrication equation for calculating the stiffness and damping coefficients. Assuming the journal is excited into a small amplitude harmonic motion with respect to the steady-state equilibrium position (*ε*_0_, *θ*_0_) within the compliant microbearing at excitation frequency Ω. The instantaneous eccentricity ratio and attitude angle may be written respectively as:(19)$$\left\{ \begin{array}{l} \varepsilon =\varepsilon_{0}+E_{0}e^{i\Omega T}\\ \theta =\theta_{0}+\Theta_{0}e^{i\Omega T} \end{array}\right.$$
where *E*_0_ and Θ_0_ are the perturbation amplitudes of eccentricity ratio and attitude angle in the complex field. Ω is the nondimensional perturbation frequency, which is defined as the ratio of journal perturbation frequency *ν* to journal rotation velocity *ω*, $i=\sqrt{-1}$.

The small perturbation of gas film thickness causes variation in the pressure field, then the dimensionless film thickness *H* and film pressure *P* that consist of steady and dynamic components can be expressed as:(20)$$\left\{ \begin{array}{l} P=P_{0}+{\tilde{P}}_{0}e^{i\Omega T}\\ H=H_{0}+{\tilde{H}}_{0}e^{i\Omega T}\\ {\tilde{H}}_{0}=E_{0}\cos(\varphi -\theta_{0})+\varepsilon_{0}\Theta_{0}\sin(\varphi -\theta_{0}) \end{array}\right.$$
where *P*_0_ is the static gas pressure and *H*_0_ is the static film thickness. ${\tilde{P}}_{0}$ and ${\tilde{H}}_{0}$ are the complex perturbation magnitudes for dynamic gas film pressure and thickness, respectively.

On substituting Equations (20) into Equation (1), the dynamic modified Reynolds equation for the effective viscosity and rarefaction effects of gas microbearing with elastic liner is derived as:(21)$$\begin{array}{l} \frac{\partial }{\partial \varphi }(QP_{0}{H_{0}}^{3}\frac{\partial {\tilde{P}}_{0}}{\partial \varphi })+\frac{\partial }{\partial \lambda }(QP_{0}{H_{0}}^{3}\frac{\partial {\tilde{P}}_{0}}{\partial \lambda })+\frac{\partial }{\partial \varphi }(Q{\tilde{P}}_{0}{H_{0}}^{3}\frac{\partial P_{0}}{\partial \varphi })+\\ \frac{\partial }{\partial \lambda }(Q{\tilde{P}}_{0}{H_{0}}^{3}\frac{\partial P_{0}}{\partial \lambda })+\frac{\partial }{\partial \varphi }(3QP_{0}{H_{0}}^{2}{\tilde{H}}_{0}\frac{\partial P_{0}}{\partial \varphi })+\frac{\partial }{\partial \lambda }(3QP_{0}{H_{0}}^{2}{\tilde{H}}_{0}\frac{\partial P_{0}}{\partial \lambda })=\\ \Lambda_{eff}\frac{\partial }{\partial \varphi }(P_{0}{\tilde{H}}_{0}+{\tilde{P}}_{0}H_{0})+i2\Lambda_{eff}\Omega (P_{0}{\tilde{H}}_{0}+{\tilde{P}}_{0}H_{0}) \end{array}$$

In reference [46], the dimensionless quantities are defined as:(22)$$\left\{ \begin{array}{l} P_{E}=\frac{\partial {\tilde{P}}_{0}}{\partial E_{0}},\\ P_{\theta }=\frac{1}{\varepsilon_{0}}\frac{\partial {\tilde{P}}_{0}}{\partial \Theta_{0}},\\ H_{E}=\frac{\partial {\tilde{H}}_{0}}{\partial E_{0}},\\ H_{\theta }=\frac{1}{\varepsilon_{0}}\frac{\partial {\tilde{H}}_{0}}{\partial \Theta_{0}} \end{array}\right.$$

Differentiating the resulting equation with respect to *E*_0_ and Θ_0_, the following dynamic Reynolds equations for elastic microbearing regarding *P_E_*, *P_θ_*, *H_E_*, and *H_θ_* are obtained:
(23)$$\begin{array}{l} \frac{\partial }{\partial \varphi }(QP_{0}H_{0}{}^{3}\frac{\partial P_{E}}{\partial \varphi })+\frac{\partial }{\partial \lambda }(QP_{0}H_{0}{}^{3}\frac{\partial P_{E}}{\partial \lambda })+\frac{\partial }{\partial \varphi }(QP_{E}H_{0}{}^{3}\frac{\partial P_{0}}{\partial \varphi })+\frac{\partial }{\partial \lambda }(QP_{E}H_{0}{}^{3}\frac{\partial P_{0}}{\partial \lambda })+\\ \frac{\partial }{\partial \varphi }(\frac{\partial Q}{\partial E_{0}}P_{0}H_{0}{}^{3}\frac{\partial {\tilde{P}}_{0}}{\partial \varphi })+\frac{\partial }{\partial \lambda }(\frac{\partial Q}{\partial E_{0}}P_{0}H_{0}{}^{3}\frac{\partial {\tilde{P}}_{0}}{\partial \lambda })+\frac{\partial }{\partial \varphi }(\frac{\partial Q}{\partial E_{0}}{\tilde{P}}_{0}H_{0}{}^{3}\frac{\partial P_{0}}{\partial \varphi })+\\ \frac{\partial }{\partial \lambda }(\frac{\partial Q}{\partial E_{0}}{\tilde{P}}_{0}H_{0}{}^{3}\frac{\partial P_{0}}{\partial \lambda })+3QP_{0}H_{0}{}^{3}\frac{\partial P_{0}}{\partial \varphi }\frac{\partial }{\partial \varphi }(\frac{H_{E}}{H_{0}})+3QP_{0}H_{0}{}^{3}\frac{\partial P_{0}}{\partial \lambda }\frac{\partial }{\partial \lambda }(\frac{H_{E}}{H_{0}})+\\ 3\frac{\partial }{\partial \varphi }(\frac{\partial Q}{\partial E_{0}}P_{0}H_{0}{}^{2}{\tilde{H}}_{0}\frac{\partial P_{0}}{\partial \varphi })+3\frac{\partial }{\partial \lambda }(\frac{\partial Q}{\partial E_{0}}P_{0}H_{0}{}^{2}{\tilde{H}}_{0}\frac{\partial P_{0}}{\partial \lambda })+3\Lambda_{eff}\frac{H_{E}}{H_{0}}\frac{\partial (P_{0}H_{0})}{\partial \varphi }=\\ \Lambda_{eff}\frac{\partial }{\partial \varphi }(P_{0}H_{E}+P_{E}H_{0})+i2\Lambda_{eff}\Omega (P_{0}H_{E}+P_{E}H_{0}) \end{array}$$
(24)$$H_{E}=\cos(\varphi -\theta_{0})$$
(25)$$\begin{array}{l} \frac{\partial }{\partial \varphi }(QP_{0}H_{0}{}^{3}\frac{\partial P_{\theta }}{\partial \varphi })+\frac{\partial }{\partial \lambda }(QP_{0}H_{0}{}^{3}\frac{\partial P_{\theta }}{\partial \lambda })+\frac{\partial }{\partial \varphi }(QP_{\theta }H_{0}{}^{3}\frac{\partial P_{0}}{\partial \varphi })+\frac{\partial }{\partial \lambda }(QP_{\theta }H_{0}{}^{3}\frac{\partial P_{0}}{\partial \lambda })+\\ \frac{\partial }{\partial \varphi }(\frac{\partial Q}{\partial \Theta_{0}}P_{0}H_{0}{}^{3}\frac{\partial {\tilde{P}}_{0}}{\partial \varphi })+\frac{\partial }{\partial \lambda }(\frac{\partial Q}{\partial \Theta_{0}}P_{0}H_{0}{}^{3}\frac{\partial {\tilde{P}}_{0}}{\partial \lambda })+\frac{\partial }{\partial \varphi }(\frac{\partial Q}{\partial \Theta_{0}}{\tilde{P}}_{0}H_{0}{}^{3}\frac{\partial P_{0}}{\partial \varphi })+\\ \frac{\partial }{\partial \lambda }(\frac{\partial Q}{\partial \Theta_{0}}{\tilde{P}}_{0}H_{0}{}^{3}\frac{\partial P_{0}}{\partial \lambda })+3QP_{0}H_{0}{}^{3}\frac{\partial P_{0}}{\partial \varphi }\frac{\partial }{\partial \varphi }(\frac{H_{\theta }}{H_{0}})+3QP_{0}H_{0}{}^{3}\frac{\partial P_{0}}{\partial \lambda }\frac{\partial }{\partial \lambda }(\frac{H_{\theta }}{H_{0}})+\\ 3\frac{\partial }{\partial \varphi }(\frac{\partial Q}{\partial \Theta_{0}}P_{0}H_{0}{}^{2}{\tilde{H}}_{0}\frac{\partial P_{0}}{\partial \varphi })+3\frac{\partial }{\partial \lambda }(\frac{\partial Q}{\partial \Theta_{0}}P_{0}H_{0}{}^{2}{\tilde{H}}_{0}\frac{\partial P_{0}}{\partial \lambda })+3\Lambda_{eff}\frac{H_{\theta }}{H_{0}}\frac{\partial (P_{0}H_{0})}{\partial \varphi }=\\ \Lambda_{eff}\frac{\partial }{\partial \varphi }(P_{0}H_{\theta }+P_{\theta }H_{0})+i2\Lambda_{eff}\Omega (P_{0}H_{\theta }+P_{\theta }H_{0}) \end{array}$$
(26)$$H_{\theta }=\sin(\varphi -\theta_{0})$$

According to the coordinate of the beginning of the lubricating film, as illustrated in Figure 1, using an iterative partial derivative algorithm, the dynamic stiffness and damping coefficients *K_ij_* and *D_ij_* can be solved from the following integral equation:(27)$$\left\{ \begin{array}{l} -\frac{R}{B}\iint_{A}P_{E}\cos\varphi d\varphi d\lambda =K_{y\varepsilon }+i\Omega D_{y\varepsilon }\\ \frac{R}{B}\iint_{A}P_{E}\sin\varphi d\varphi d\lambda =K_{x\varepsilon }+i\Omega D_{x\varepsilon }\\ -\frac{R}{B}\iint_{A}P_{\theta }\cos\varphi d\varphi d\lambda =K_{y\theta }+i\Omega D_{y\theta }\\ \frac{R}{B}\iint_{A}P_{\theta }\sin\varphi d\varphi d\lambda =K_{x\theta }+i\Omega D_{x\theta } \end{array}\right.$$

Using a transformation matrix *A*, the dynamic stiffness and damping properties of microbearings can be converted into the Cartesian coordinate system.
(28)$$\left\{ \begin{array}{l} \left(K_{ij}\right)=\left(\begin{array}{cc} K_{xx} & K_{xy}\\ K_{yx} & K_{yy} \end{array}\right)=A\left(\begin{array}{cc} K_{x\varepsilon } & K_{x\theta }\\ K_{y\varepsilon } & K_{y\theta } \end{array}\right)\\ \left(D_{ij}\right)=\left(\begin{array}{cc} D_{xx} & D_{xy}\\ D_{yx} & D_{yy} \end{array}\right)=A\left(\begin{array}{cc} D_{x\varepsilon } & D_{x\theta }\\ D_{y\varepsilon } & D_{y\theta } \end{array}\right) \end{array}\right.,(i,j=x,y),A=\left(\begin{array}{cc} -\sin\theta_{0} & -\cos\theta_{0}\\ \cos\theta_{0} & -\sin\theta_{0} \end{array}\right)$$

This paper shows, theoretically, the importance of the rarefaction effect, elastic deformation of the bearing bounding surface and effective viscosity on rarefied gas flow in ultra-thin bearings. To guarantee the stability of micro-rotating machinery, it is highly desirable to achieve more accurate microbearing characteristics appearing in microbearing–rotor system may be crucial to the design of micro-fluid machines in microelectromechanical (MEMS) systems.

## 3. Elastic Deformation and Fluid Film Thickness

The elastic microbearing operating under nominal loads is required to maintain an appropriate gas film profile for supporting the journal. The shape and distribution of gas film thickness are directly affected by the elastic deformation of the bearing liner and vice versa. The micro-elasto-aerodynamic lubrication is a fluid-structure interaction problem and the effects of effective viscosity and bearing deformation on film pressure should be taken into account in each iterative solution procedure. The main obstacle appears to be the convergence criterion of pressure; deformation and variable viscosity must be satisfied simultaneously within the specified tolerance. The partial derivative method is used to solve the modified Reynolds equation governing the generation of pressure for compressible fluids and finite element technique is employed to solve the three-dimensional linear elasticity equation for the displacement field of bearing liner.

In the conventional EHL analysis, the bearing surface elastic deformation is assumed to be a semi-infinite elastic solid with elastic modulus *E* and Poisson ratio *υ*. In order to reflect the actual viscoelastic behaviors, the bearing shell is spread along the circumferential direction and is divided by a network of eight-noded hexahedral linear isoparametric elements, as shown schematically in Figure 2. The elastic distortion of the bounding solids under film pressure is very small and the deformation can be considered to be varying linearly with load. The corresponding element stiffness matrix can be built up:(29)$$K^{e}={∭}_{V}B^{T}DBdxdydz$$
(30)$$B=\left[B_{1}\;B_{2}\;B_{3}\;\ldots \;B_{8}\right],\;B_{i}=\left(\begin{array}{llllll} \frac{\partial N_{i}}{\partial x} & 0 & 0 & \frac{\partial N_{i}}{\partial y} & 0 & \frac{\partial N_{i}}{\partial z}\\ 0 & \frac{\partial N_{i}}{\partial y} & 0 & \frac{\partial N_{i}}{\partial x} & \frac{\partial N_{i}}{\partial z} & 0\\ 0 & 0 & \frac{\partial N_{i}}{\partial z} & 0 & \frac{\partial N_{i}}{\partial y} & \frac{\partial N_{i}}{\partial x} \end{array}\right),\;(i=1,2,3\ldots ,8)$$
(31)$$D=\frac{E}{\left(1+\upsilon )(1-2\upsilon \right)}\left(\begin{array}{llllll} 1-\upsilon & 0 & 0 & 0 & 0 & 0\\ 0 & 1-\upsilon & 0 & 0 & 0 & 0\\ 0 & 0 & 1-\upsilon & 0 & 0 & 0\\ 0 & 0 & 0 & \frac{1-2\upsilon }{2} & 0 & 0\\ 0 & 0 & 0 & 0 & \frac{1-2\upsilon }{2} & 0\\ 0 & 0 & 0 & 0 & 0 & \frac{1-2\upsilon }{2} \end{array}\right)$$
where *N* is the shape function.

The element stiffness matrix is easy to assemble the global stiffness matrix of microbearing by direct stiffness method. The finite element formulation of this model is obtained by applying the principle of minimum potential energy. Compliance matrix *C* is established to determine the radial deformation of a given sequence of nodes on the bearing surface caused by the gas pressure within the film region as follows:(32)$$C=K^{-1}$$
(33)$$\delta_{t}=\sum C\cdot P$$
where *δ_t_* is the radial distortion matrix of all nodes on the inner surface of microbearing under gas film pressure matrix *P*.

Using the nodal displacements of microbearing surface, the elastic deformation is computed through the compliance matrix which is derived from the finite element model and the film thickness is modified. The gas film thickness *H* is made up of two parts for the elastic gas journal microbearing system:(34)$$H=H_{0}+\delta_{t}=1+\varepsilon \cos(\varphi -\theta )+\delta_{t}$$
where *H*_0_ represents the film profile in the flow region of rigid bearing, *δ_t_* is the elastic deformation of bearing liner due to the gas film pressure and effective lubricant viscosity dependence of the Knudsen number.

The corresponding boundary conditions are described as follows:

The inner surface of the bearing liner, namely, the upper surface of the model is subjected to continuously distributed film pressure. The pressure load is transformed into equivalent loads acting on each node of the upper surface.

The outer surface of the bearing liner is enclosed in a rigid housing. Hence, the nodal displacement components for the outer surface of the bearing liner are restrained from moving and are made zero.

The elastic bearing liner is a closed cylindrical structure in the circumferential direction and the starting and ending cross-sections in the finite element model of microbearing are the same so that the radial displacements of liner-housing interface are identical in the simulation.

## 4. Results and Discussion

In this section, the results for rigid and deformable gas journal microbearings with or without the effective viscosity operating in laminar and isothermal conditions are compared and discussed in Figures 4–11. The static and dynamic bearing performance in terms of load capacity, friction coefficient, direct stiffness, and damping coefficients are given for various eccentricity ratios (0.1–0.8) and the modulus of elasticity (5–200GPa). The geometry parameters of the microbearing are selected: *c* = 1 µm, *R* = 1 mm, *B* = 200 µm, *p_a_* = 1.033 × 10^5^ N/m^2^, and the aspect ratio *B*/*D* = 0.1.

In order to verify the algorithm and computer program developed in the present article, as shown in Figure 3, the dimensionless gas pressures computed by the authors are compared with the numerical predictions reported by Zhang et al. for *R* = 2.0 mm, *B* = 0.4 mm, *p_a_* = 1.033 × 10^5^ N/m^2^, Λ = 2.4 in Ref. [30]. It is confirmed that both results are in good agreement and showed the correctness of the mathematical model.

### 4.1. Steady-State Characteristics

Figure 4 presents the effects of elastic deformation and effective viscosity on the gas film pressure distribution at the mid plane for different values of the bearing number Λ. It is observed that increasing the value of the bearing number from Λ = 3 up to 100 increases the magnitude of the maximum pressure. In comparison with the rigid bearing case, the elastic deformation of bearing liner reduces the size of maximum film pressure and a bimodal pressure exists along the sliding direction, especially for higher values of the bearing number. This is because the more obvious aerodynamic effect results in higher local pressure peaks and the gas film thickness has two converging-diverging profiles unlike the rigid bearing liner. The effective gas viscosity will further decrease the pressure in the microbearing with the elastic liner.

Figure 5 describes the variation of the load carrying capacity *C_L_* with eccentricity ratio *ε* for different values of r of elasticity *E* of microbearing liner, respectively. It is seen that the load capacity becomes larger and larger with increasing eccentricity ratio and the gas rarefaction effect decreases all components of load capacities. As the elastic modulus increases from 5 to 200 GPa, the non-dimensional load-carrying capacity of the gas-lubricated journal microbearing increases gradually for the elastic cases when the eccentricity ratio *ε* < 0.7, while *C_L_* decreases as *E* increases at high values of *ε* and the increases in *C_L_* are small. The reason behind this is that the load capacity can be attained by integrating the pressure distribution over the film region and the pressure integral areas are larger for elastic bearing liners. Further, the magnitudes of load-carrying capacities are higher in the constant viscosity cases than the ones of the effective gas viscosity with an increase in *ε*.

The variation of the friction coefficient with the eccentricity ratio for rigid and elastic liners are shown in Figure 6. The curves indicate that the friction coefficient increases with the increase of both the eccentricity ratio and modulus of elasticity for *ε* < 0.6, while it decreases with an increase of the elastic modulus of liner material when the journal operates at higher eccentricity ratios. The reason for the phenomenon is that the elastic deformation of the microbearing surface increases the minimum gas film clearance, where the gas flow is less restricted and the increase of the lubricant shear stress is more significant over the bearing surface. Meanwhile, the effective viscosity of the gas lubricant film with rarefaction decreases the friction coefficient along the film thickness.

Figure 7 shows the relationship between the steady load-carrying capacity *C_L_* and the bearing number Λ at different bearing flexibility combining with the effective viscosity in ultra-thin films. It can be found that the non-dimensional load-carrying capacity of gas microbearing almost increases linearly with the increase in the bearing number, which corresponds to the higher journal speed around the lubricated surface and compressibility effect in gas microbearing. The load capacity increases as the bearing flexibility increases, this is a consequence of enhancing the micro-elastoaerodynamic effects in the elastic cases that weaken the gaseous rarefaction effect in microbearing to some extent. The non-dimensional load capacities for an elastic bearing with a lubricant of constant viscosity are higher than that for the corresponding effective viscosity cases, and the more elastic the bearing liner, the more obvious the increase in the *C_L_* at a larger bearing number.

The variations of friction coefficient with bearing number Λ for different bearing shell flexibility is shown in Figure 8. The bearing number makes a clear difference in the dimensionless friction coefficient which was expected because in aerodynamic journal microbearings, friction comes from viscous shear stress of the rarefied gas flow and the higher bearing number means the higher rotating speed and more viscous shear stress. With the increment of bearing flexibility, the friction coefficients increase, which is attributed to the increase in gas film thickness with decreasing modulus of elasticity *E*. This indicates that the increase in the friction coefficients is relatively apparent compared with the increase in the load-carrying capacity when the elastic deformations of the bearing liner is considered. Moreover, the friction coefficients of the journal surface taking into account the effective viscosity of the lubricant are lower than that predicted by the cases of constant viscosity.

### 4.2. Dynamic Stiffness and Damping Coefficients

Figure 9 depicts the variation of dynamic stiffness and damping coefficients as a function of dimensionless perturbation frequency Ω for different values of elastic moduli by considering the effective viscosity of the rarefied gas lubricant. It is observed that the effect of perturbation frequency is to increase the direct stiffness coefficients for both cases of rigid and elastic microbearings, and the *K_yy_* is slightly larger than *K_xx_* on account of the lubricating gas film supports the rotor weight in the vertical direction. From the figures, it is also noticed that the increase in stiffness coefficients are more accentuated for low elastic moduli as compared to rigid bearing liner cases. The plausible reason for this is that, for the elastic bearings, a small increase in film thickness resulting from the elastic deformation of bearing liner can cause an increase in the inflow gas inside the small gap. All the direct dynamic damping coefficients of gas-lubricated journal microbearing decrease as the perturbation frequency increases. As the elastic modulus decreases, which indicates that the flexibility of the bearing shell is higher, the direct damping coefficients are lower than that of the rigid bearing liner. When the effective viscosity of the lubricant is considered, the direct stiffness coefficients become smaller and the direct damping coefficients increase with increasing perturbation frequency compared to the constant viscosity cases at higher values of Ω for rigid and elastic microbearings.

Figure 10 illustrates the variation of the relative difference in the direct stiffness and damping coefficients of microbearing with the eccentricity ratio *ε* when considering and not considering the effective gas viscosity for Ω = 3.5 and Λ = 80. It is seen that all the direct stiffness coefficients increase and all the direct damping coefficients decrease slightly for increasing *ε*. The effect of elastic deformation is notable for higher eccentricity ratios and it is found that the stiffness coefficients increase gradually whereas the direct damping coefficients decrease with the increase in modulus of elasticity at the same rotor eccentricity ratio. The reason is attributed to the decrease of the gas film thickness leads to the increase of the Knudsen number, which results in the augmentation of the Poiseuille term in the perturbed Reynolds equation of gas films. Furthermore, the direct stiffness coefficients for microbearings with a lubricant of effective viscosity are always lower than that for constant viscosity cases while the trends are opposite in the direct damping coefficients.

Figure 11 display comparisons of the dynamic characteristics with bearing number Λ at different moduli of elasticity of the bearing shell material for fixed values of Ω = 3.5 and *ε* = 0.7. Increasing values of bearing number produce the increasing effect in fluid film stiffness coefficients, while the direct damping coefficients first increases and then begins to fall when Λ > 30. As compared to a rigid microbearing, the direct stiffness coefficients increase and direct damping coefficients decrease with the decrease of the elastic modulus. This is because the ultra-thin gas film is stiffened at higher bearing numbers and can prevent the dominant energy dissipation in such narrow gaps. In many ultra-thin gas lubrication applications, it is beneficial that the damping coefficient has smaller values. The effective viscosity effect in the gas film increases the direct damping coefficients and the stiffness coefficients of microbearing become smaller than those of the microbearing lubricated with the gas lubricant of constant viscosity. This reveals that if the gas viscosity is modified to include gaseous rarefaction, the effective gas viscosity in thin films serves to increase damping.

## 5. Conclusions

Based upon the effective viscosity model proposed by Veijola for rarefied gas flow, the modified Reynolds equation coupled with the elasticity equation is solved numerically by applying the partial derivative method, relaxation iterative scheme, and finite element formulation. The effects of gas rarefaction and the effective viscosity of lubricant considerably affect the micro-elasto-aerodynamic microbearing characteristics and design of microbearing−rotor system. Some significant results are summarized as follows:

(1)The presence of the elastic bearing liner is seen to decrease the maximum gas film pressure around the local pressure peak region and the elastic journal microbearings with effective viscosity carry less load as compared to the cases for rigid bearings lubricated with constant viscosity gas lubricant.(2)The load-carrying capacity marginally increases in the range of high eccentricity ratio and bearing number for elastic bearing liner because of the larger integral area of film pressure. The effects of bearing shell flexibility provide an increase in the friction coefficient compared to the rigid bearing, whereas the influence of effective viscosity yield a reversed trend.(3)Improving the perturbation frequency, bearing number, and eccentricity ratio contributes to the increase in the direct stiffness coefficients while the direct damping coefficients become smaller and smaller with increasing dimensionless perturbation frequency and eccentricity ratios.(4)As the elastic moduli decreases, the direct stiffness coefficients increase while the direct damping coefficients decrease. The effective viscosity of gas lubricant decreases the direct stiffness coefficients of microbearings and the direct damping coefficients are larger than those predicted by the constant viscosity lubricant.

## Figures and Tables

**Figure 1 micromachines-10-00657-f001:**
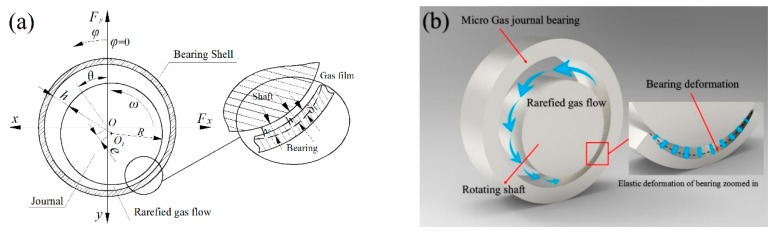
(**a**) Schematic and (**b**) 3-dimensional graphical representation of a gas lubricated microbearing considering elastic deformation of the bearing surface.

**Figure 2 micromachines-10-00657-f002:**
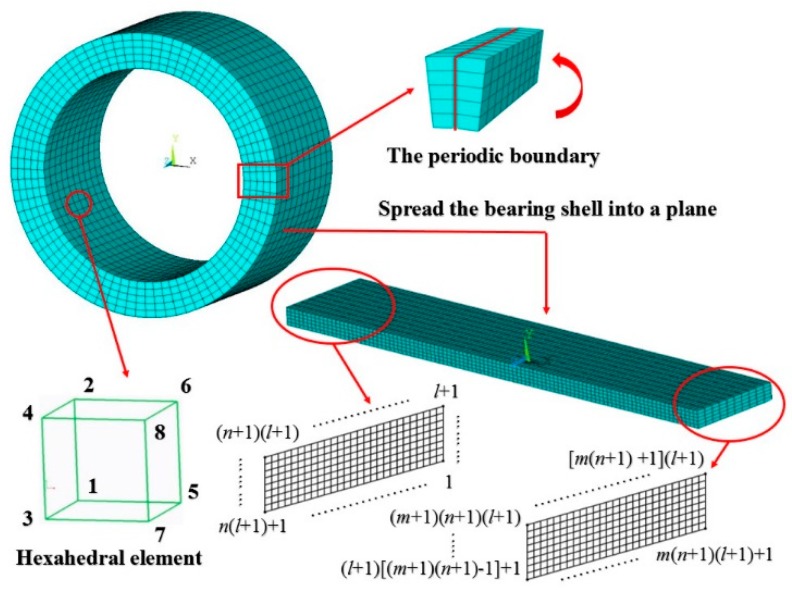
The entity unit and finite element model of gas microbearing.

**Figure 3 micromachines-10-00657-f003:**
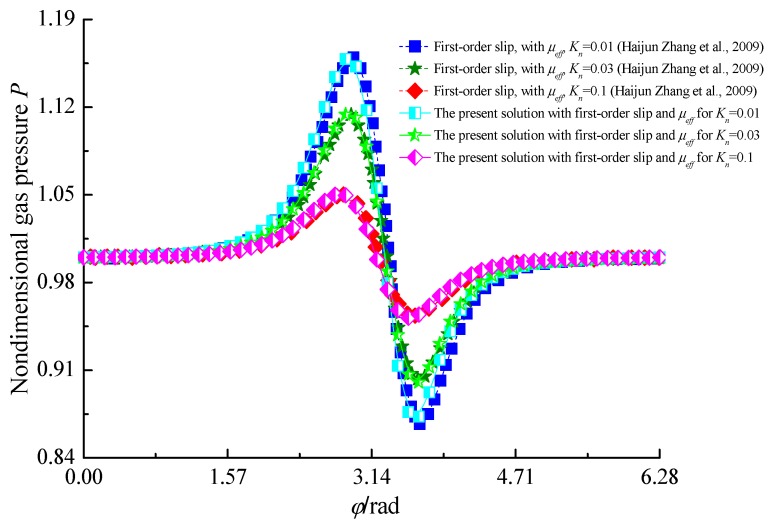
Comparison of dimensionless gas pressure at the mid-plane with Zhang et al. [20].

**Figure 4 micromachines-10-00657-f004:**
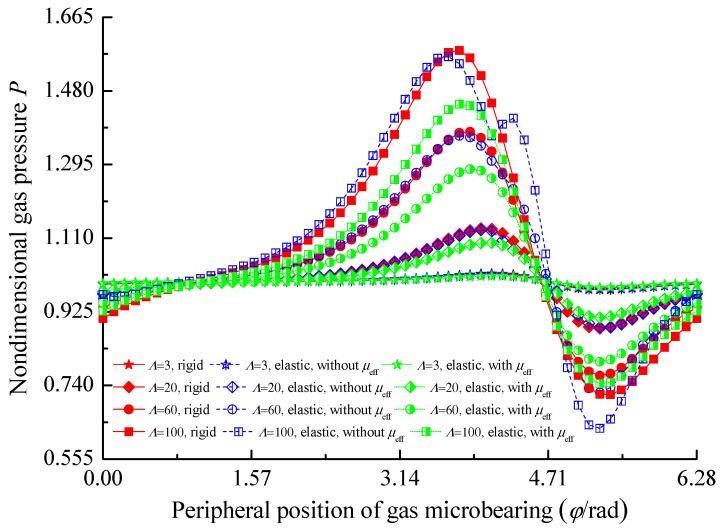
Pressure distributions of the gas microbearing for different bearing numbers (*ε* = 0.6, *E* = 300 GPa, *υ* = 0.3).

**Figure 5 micromachines-10-00657-f005:**
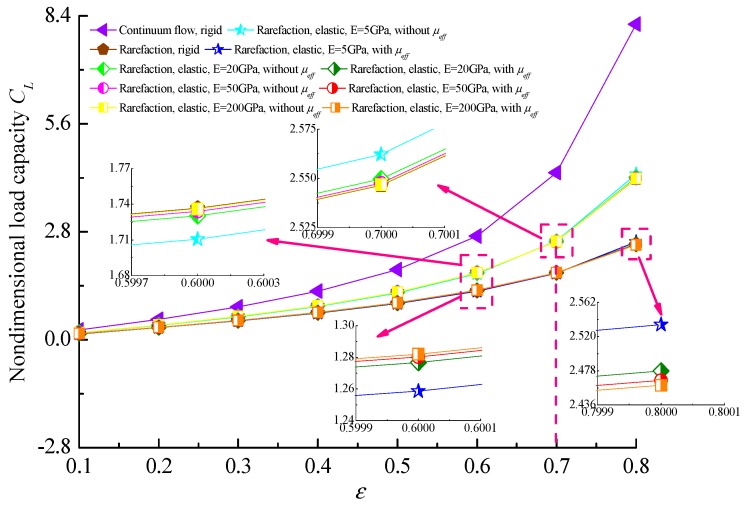
Non-dimensional load capacity versus the eccentricity ratio for different elastic moduli at Λ = 30 and *υ* = 0.3.

**Figure 6 micromachines-10-00657-f006:**
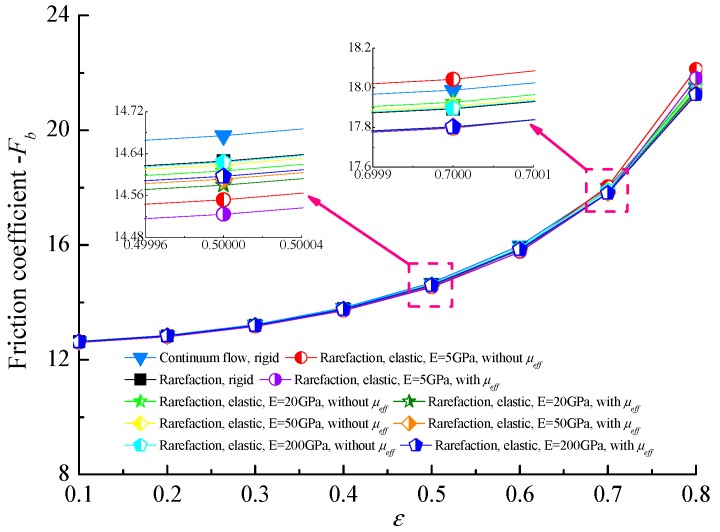
Relationships between the static friction coefficient and the eccentricity ratios for different elastic moduli at Λ = 30 and *υ* = 0.3.

**Figure 7 micromachines-10-00657-f007:**
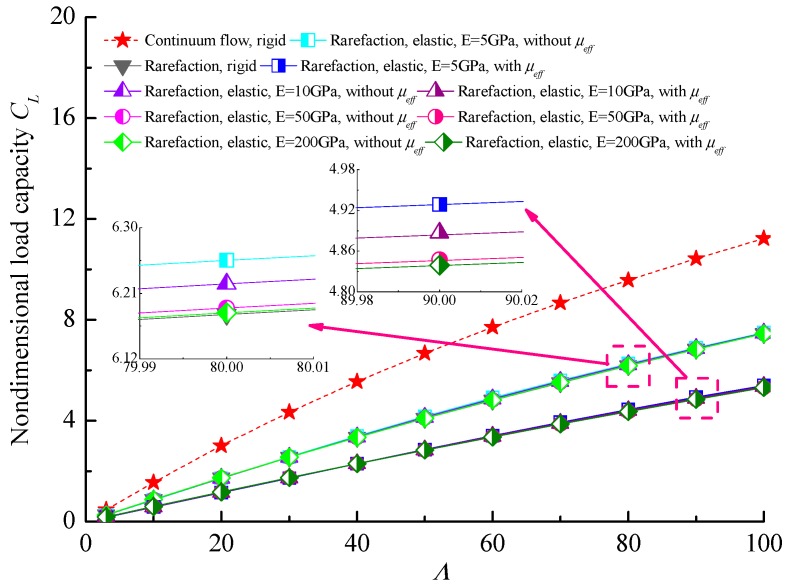
Effect of bearing number on the non-dimensional load capacity for different elastic moduli with and without effective viscosity (*ε* = 0.7, *υ* = 0.3).

**Figure 8 micromachines-10-00657-f008:**
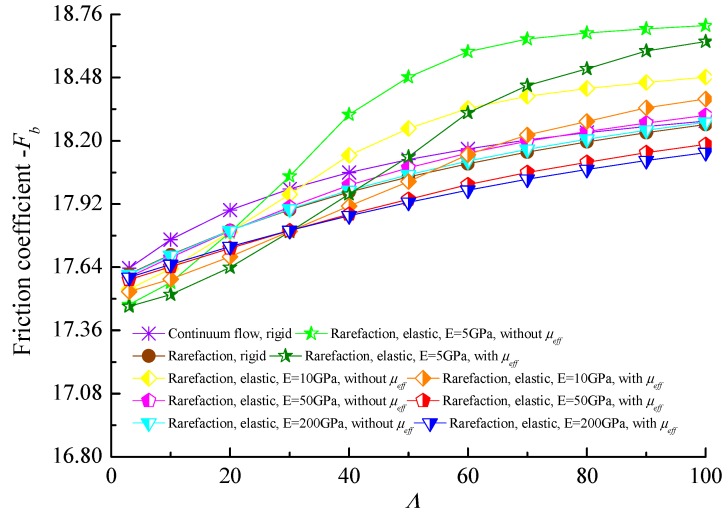
Friction coefficient against the bearing number with effective and constant viscosity for the different elastic moduli of microbearing material (*ε* = 0.7, *υ* = 0.3).

**Figure 9 micromachines-10-00657-f009:**
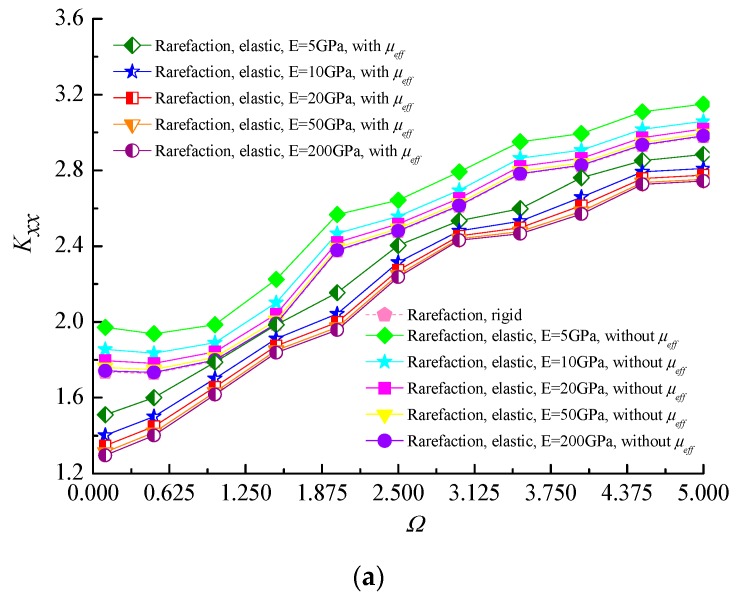
Variation of dynamic stiffness and damping coefficients with the perturbation frequency for different moduli of elasticity of liner material. (**a**) *K_xx_* vs. Ω; (**b**) *K_yy_* vs. Ω; (**c**) *D_xx_* vs. Ω; (**d**) *D_yy_* vs. Ω (*ε* = 0.7, Λ = 80, *υ* = 0.3).

**Figure 10 micromachines-10-00657-f010:**
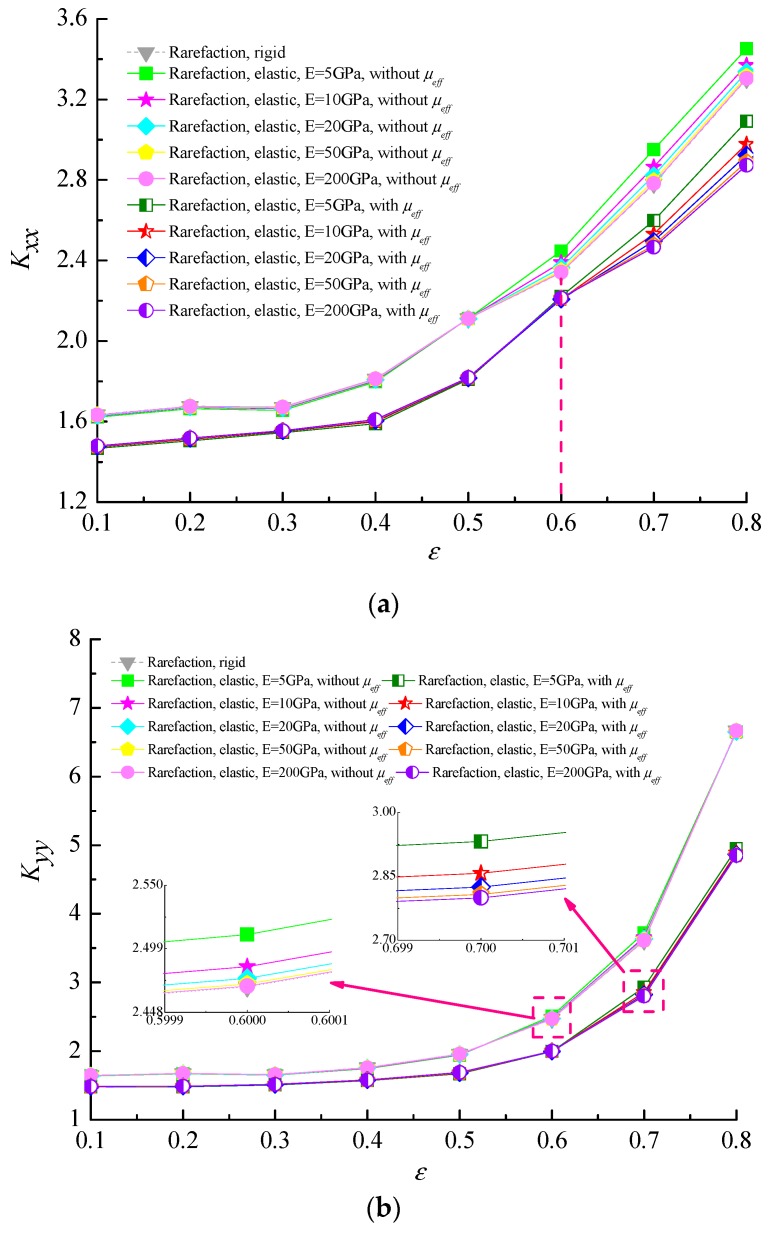
Effect of eccentricity ratio on dynamic stiffness and damping coefficients for variable and constant viscosity of lubricant with different elastic moduli. (**a**) *K_xx_* vs. *ε*; (**b**) *K_yy_* vs. *ε*; (**c**) *D_xx_* vs. *ε*; (**d**) *D_yy_* vs. *ε* (Ω = 3.5, Λ = 80, *υ* = 0.3).

**Figure 11 micromachines-10-00657-f011:**
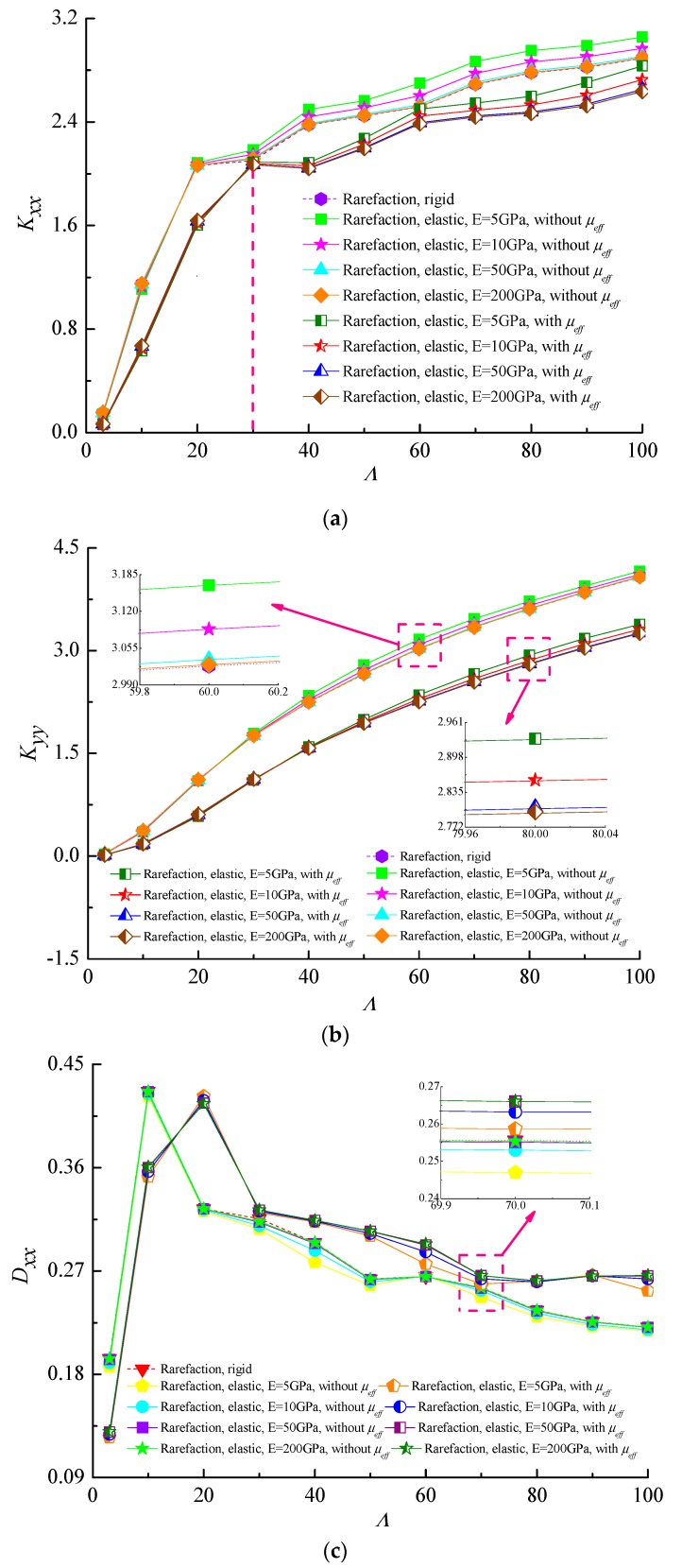
Dynamic stiffness and damping coefficients as functions of bearing number for different values of elastic moduli when considering or not considering the effective viscosity. (**a**) *K_xx_* vs. Λ; (**b**) *K_yy_* vs. Λ; (**c**) *D_xx_* vs. Λ; (**d**) *D_yy_* vs. Λ (Ω = 3.5, *ε* = 0.7, *υ* = 0.3).

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
