# Peer review of "Combined Effect of Rarefaction and Effective Viscosity on Micro-Elasto-Aerodynamic Lubrication Performance of Gas Microbearings"

_micromachines, 2019, doi:10.3390/mi10100657_

Round 1

Reviewer 1 Report

The authors compare an enhanced EHD model for micro gas bearing performance with a traditional aproach considering a rigid bearing shell and constant viscosity. 

The authors highlight the differences in the bearing perfirmance characteristics, caused by the effective viscosity approach and the deformable bearing shell.

I find the analysis robust and the related results usefull for the scientific community. There are only some minor revisions:

1) In the abstract, the reader may be confused as the authors imply from one hand  that the variation trend is inverse (increase) in the load capacity. froction coeficient and stifffness coeficient at high eccentricity ratios and bearing numbers, when bearing flexibility is assumed, but, from the other hand it is declared that bearing flexibility decreases the pressure distribution inside the bearing. Please clarify this in abstract.

2) Most of the Figures 3 to 17 have much information (this is good), and I would expect high resolution. If the authors consider practically feasible to increase resolution to e.g. 600dpi (probably it is 300dpi now), the Figures will gain more quality and the article better appearance.

Author Response

Dear Reviewer:

We are truly grateful to your comments and suggestions concerning our manuscript entitled “Combined effect of rarefaction and effective viscosity on micro-elasto-aerodynamic lubrication performance of gas microbearings” (Manuscript ID: micromachines-583226). The comments and suggestions are very valuable and helpful for revising and improving our paper. We have carefully addressed all the comments and revised the manuscript accordingly, and corresponding content are renewed in red color. Attached please find the revised version, which we would like to submit for your kind consideration.

Thank you very much for your time and consideration, and look forward to hearing from you soon.

Best regards,

Sincerely yours,

Yao Wu

Reviewer 2 Report

The present manuscript deals with computational study of the fluid rarefaction and viscosity effects on the lubrication performance of gas micro bearings. The authors state that the fluid compressibility, rarefaction and  viscosity effects would be important on the lubrication performance, but they need to mention about the basic physics underlying in these effects, and no detailed results were provided in the present work. More fatal issue is that there is no clear validation of the present simulation results. The present paper seems too lengthy and also the conclusions should  be provided with the results which are well matched with the research purposes as stated in the introduction. 

Author Response

Dear Reviewer:

Thank you very much for your letter and for the kind and instructive comments and suggestions concerning our manuscript entitled “Combined Effect of Rarefaction and Effective Viscosity on Micro-elasto-aerodynamic Lubrication Performance of Gas Microbearings” (ID: micromachines-583226). Those comments are all valuable and very helpful for revising and improving our paper, as well as the important guiding significance to our future research. Accoring to the beneficial suggestions provided in your letter, We have studied comments carefully and have made correction which we hope meet with approval. Revised portion are marked in a different color (red) in the manuscript.

Sincerely,

Yao Wu
